# Determination of Selected Harmful Substances in Baby Diapers Available on the South African Market

**DOI:** 10.3390/ijerph20021023

**Published:** 2023-01-06

**Authors:** Pardon Nyamukamba, Zethu Mququ, Sandile Nkosi, Shamil Isaacs

**Affiliations:** Technology Station Clothing and Textiles, Symphony Way, Bellville, Cape Town 7535, South Africa

**Keywords:** diapers, heavy metals, formaldehyde, Oeko-Tex limits

## Abstract

Baby diaper rash is a common problem, especially allergic contact dermatitis, which could be due to heavy metals, pH, formaldehyde, or allergens in the diapers. This study reports on the determination of formaldehyde, heavy metals (Pb, As, Co, Cr, Ni, Cu, Zn, Mn, Sr, Fe, and Cd), and pH in diapers purchased from low-, medium-, and high-end stores. Inductively coupled plasma was used to determine the concentrations of heavy metals after extraction using artificial urine and artificial sweat. All heavy metals were found in all diapers except Sr, which was not found in sample M7. All samples had concentrations of heavy metals within the Oeko-Tex limits, except samples H2 and L2, whose Ni concentrations were above permissible limits. Fifty percent of diapers had a formaldehyde concentration above the Oeko-Tex recommended limits. The highest formaldehyde concentration of 17.62 mg/kg was found in diaper M2 and the lowest (ca. 10.4 mg/kg) in H1. All samples had pH values in the alkaline region, with only five samples having pH values within the recommended limits. The research concluded that the diaper rash experienced by some babies, among other factors, could be due to high alkaline skin pH and formaldehyde levels, which are higher than the Oeko-Tex recommended limits.

## 1. Introduction

Before the introduction of super-absorbent polymer (SAP), cloth diapers were used, and these were made from cotton for soft stuffing and to protect the baby. The major disadvantages of cloth diapers are that they must be changed frequently because of their low absorbency and that they require a lot of water and detergent for cleaning [1]. The introduction of SAP enabled the production of disposable baby diapers (DBD) with high absorbency and reduced weight. SAP can absorb and retain large volumes of liquids, and it has been reported that a kilogram of SAP is capable of absorbing close to 420 L of water [2]. It has been reported that SAP in granular form can raise the retention capacity of DBD to absorb and retain a liquid hundredfold their weight [3].

Almost all disposable diapers have SAP, a crystal-like substance also known as sodium polyacrylate, which when exposed to water absorbs it through osmosis and traps it in the diaper to form a gel-like substance. The absorbed liquids are trapped within the gel structure so that the pressure due to the baby’s weight will not release the liquids. This factor is a paramount performance and functional feature of DBD. This polymer has been modified to counteract the effects of salts that are found in urine. A schematic representation of a disposable diaper is shown in Figure 1.

Since the 1990s, there has been an increase in the use of single-use disposable diapers, and it is estimated that a baby can use between 3800 to 4800 disposable diapers before the age of toilet training [4], but there have been some cases of children developing rash after the use of certain disposable diapers. The prevalence of rash and what causes it due to the use of diapers is unknown and has prompted studies to determine harmful substances in diapers. A few studies have been done on children’s diapers and women’s sanitary pads to determine the concentrations of heavy metals and other harmful substances [5,6], but there are still insufficient studies on what causes diaper dermatitis, even if there have been improvements in the properties of diapers.

Diaper dermatitis is a very common skin disease in babies, and it has been reported that the prevalence of diaper dermatitis is roughly between 7% and 50%, but it depends on hygiene practices and country, although not all cases are reported by either parents or doctors [7]. There are different types of diaper dermatitis: (i) irritant dermatitis, which is the most common form and is caused by high alkaline skin pH, skin moisture, mixing of stools and urine, or even friction between diaper and skin [8,9]; (ii) infectious dermatitis [10]; and (iii) inflammatory dermatitis, which is less common and includes allergic contact dermatitis caused by certain components in diapers [11].

Babies have soft, delicate skin that can be easily affected. Therefore, the determination of harmful substances, such as formaldehyde and heavy metals, is quite important, since they will be in direct contact with the skin. Most diapers usually have colors from dyes, and some of these dyes may contain heavy metals that can cause allergic reactions in babies. In a study done by Alberta et al. (2005), the dyes in diapers were found to cause diaper rash in babies, and it occurred only in places where the skin was in direct contact with the dyed part of the diaper. The symptoms were found to improve significantly with the use of dye-free diapers [6]. In another study, out of 667 children suspected of having allergic contact dermatitis, and 431 children with atopic dermatitis due to dyes, 4.6% were found to be sensitive to at least one dye [12]. Wirantar et al. (2019) studied the effect of diapers on skin pH, and they found that pH values increase significantly in the area covered by the diaper compared to the uncovered area [13]. 

The purpose of this study was to determine the pH and levels of formaldehyde and heavy metals in children’s disposable diapers found on the South African market. To the best of our knowledge, no data were available on the levels of formaldehyde and heavy metals in diapers at the time of the study.

## 2. Materials and Methods

### 2.1. Materials

The analytical reagents used in this study—albumin powder, creatinine, potassium chloride, sodium chloride, sodium phosphate (monobasic/monohydrate), urea, and nitric acid—were obtained from the Merck Company (Germany). Twenty different kinds of baby diapers used in South Africa, both locally manufactured and imported from China, Poland, and Malaysia, were procured from Cape Town. Table 1 shows the given sample codes, description, the countries where the diapers were manufactured, color of the diaper section in direct contact with skin. 

### 2.2. pH Determination

The pH was determined using a modified version of the method developed by the International Organization for Standardization (ISO 3071:2005), as reported by Nyamukamba et al. [14]. About 2.0 g of the diapers were cut into small pieces for effective wetting of the samples and put in a volumetric flask containing 100 mL of water (pH 7.0, 27 °C). This was shaken for about 2 h, after which 30 mL of the solution was extracted into a beaker and stirred with a pH electrode. The second and third extracts were put into beakers and then stirred, followed by pH measurements.

### 2.3. Extractable Heavy Metals Determination

In order to find the levels of extractable heavy metals, the diaper samples were extracted with an artificial urine solution and artificial sweat.

#### 2.3.1. Extraction Using Artificial Urine

About 2.0 g of the cut diapers was put in a flask containing 50 mL of artificial urine solution that was prepared using a modified version of the method by Brian and Shmaefsky [15]. About 24.27 g of urea crystals were dissolved in a liter of distilled water, followed by addition of 10 g of sodium chloride, 6 g of potassium chloride, and 6.4 g of sodium phosphate. This was then mixed until a clear solution was obtained. The pH was adjusted so that it was between 5 and 7 for normal urine, followed by the addition of 2.67 g of creatine and 0.067 g of albumin.

#### 2.3.2. Extraction Using Artificial Sweat Solution

About 2.0 g of the cut diapers was put into a flask containing 50 mL of artificial sweat solution that was prepared using the standard procedure reported in ISO 3160/2 by dissolving 20 g of sodium chloride, 17.5 g of ammonium chloride, 5 g of glacial acetic acid, and 15 g of lactic acid in a liter of deionized water and adjusting the pH to 4.7 using NaOH solution. The diapers were shaken for 24 h, followed by filtering, and finally the filtrate was analyzed by ICP-MS.

### 2.4. Inductively Coupled Plasma-Mass Spectrometry (ICP-MS) Analysis

The heavy metal concentration in the extracted samples was measured using a Spectro Arcos ICP-OES instrument equipped with a side-on plasma interface. A four-channel peristaltic pump that provides a segmented flow was used for sample inlet. Before analysis of the next sample, a 10% solution of HNO_3_ was used for washing between sample analyses. The ICP-OES measurement conditions were optimized before analysis, and the following conditions were used: plasma power, 1400 W; pump speed, 30 rpm; coolant flow, 14.00 L/min; auxiliary flow, 2.10 L/min; and nebulizer flow, 0.80 L/min.

## 3. Results and Discussion

### 3.1. FTIR Analysis

Fourier transform infrared spectroscopy (FTIR) analysis was done on an FTIR-ATR Spectrum Two supplied by PerkinElmer to determine the characteristic bonds of the SAP used in the diaper samples under study. Generally, the major peak characteristics of the expected functional groups in the SAP could be seen in all the spectra of the diapers under investigation, which agrees with what has been reported in the literature [16,17,18]. Some minor differences might be observed as different materials are available on the market to make the SAP, such as polyacrylate, polyacrylamide copolymer, ethylene maleic anhydride copolymer, and cross-linked carboxymethylcellulose, among others [1]. Figure 2a,b shows the structures of two SAPS, acrylic and cellulose-based SAP prepared via direct cross-linking of sodium carboxymethyl cellulose, respectively [19].

The FTIR spectra of diapers from high-end shops, low-end shops, and medium-end shops are shown in Figure 3A, Figure 3B, and Figure 3C, respectively. In the spectra of all the diapers under study, the absorption bands around 2935 cm^−1^ correspond to the stretching vibrations of the -CH_2_ groups, and those around 1700 cm^−1^ and 1454 cm^−1^ are due to the stretching vibrations of the carbonyl groups (C=O) and bending vibrations of the -CH_2_ groups, respectively [1,16,17]. Sample H1 is the only sample that has a peak around 3258 cm^−1^, which is broad and is related to the stretching vibrations of the OH groups [18]. It can be observed from all the spectra that the positions of the peaks were similar for all the SAPs, which suggests that there were no major structural differences in the SAPs except for sample H3. The most intense and sharp peaks were observed from the spectrum of the H1 sample. The spectrum of H3 shows very small peaks around 2900 cm^−1^ compared to other spectra, but it shows intense peaks around 1223 cm^−1^ and 1134 cm^−1^. These differences could be due to the differences in the content of carboxylate groups and in the raw materials used in the synthesis of SAP. 

### 3.2. Harmful Substances

According to Bender and Faergemann (2017), some of the compounds that frequently cause allergic contact dermatitis include fragrances (e.g., eugenol, cinnamal), heavy metals (e.g., nickel and chromium), preservatives (e.g., formaldehyde), rubber chemicals (e.g., thiuram compounds), plastic and glue chemicals (e.g., acrylics and epoxy compounds), and textile and hair dyes (e.g., toluene-2,5-diamine) [8].

#### 3.2.1. Formaldehyde

The concentrations of formaldehyde in DBD from low-end, medium-end, and high-end market stores are shown in Table 2. The highest concentration (ca. 17.617 mg/kg) was found in the diapers M2, M5, and M8 purchased from medium-end stores, and they were all above the recommended limit of 16 mg/kg. The lowest concentration (ca. 10.438 mg/kg) among all the DBD under study was found in sample H1, bought from high-end stores. The levels of formaldehyde in DBD from high-end stores followed the trend H2 > H5 > H3 > H4 > H1, those from medium-end stores followed the trend M5 = M2 = M8 > M7 > M3 = M6 = M1 > M10 > M9 > M4, and those from low-end stores followed the trend L2 > L1 > L3 > L4 > L5. Out of the five DBD from high-end stores, only one sample had a concentration that was above the recommended limit, meaning that 80% of the diapers do not pose any risk to babies due to formaldehyde. Seven out of the 10 DBD purchased from medium-end stores had a formaldehyde concentration that exceeded the Oeko-Tex recommended limit, implying that only 30% of these diapers do not pose a risk to babies due to formaldehyde. Among the diapers bought from low-end stores, two out of the five DBD had a formaldehyde concentration above the Oeko-Tex recommended limit, which means that 60% of the diapers do not pose a risk to babies due to formaldehyde. Generally, about 50% of all the diapers under study had a formaldehyde concentration above and the other 50% within the Oeko-Tex recommended limit. This implies that there are 50% chances of buying a diaper that has formaldehyde concentrations that are not within the Oeko-Tex recommended limit. It is very important for formaldehyde to be within the limits for the safety of the babies, since formaldehyde can cause skin irritation and skin allergies and is carcinogenic to humans [20,21]. It has also been found that formaldehyde causes irritation of the mucous membrane and upper respiratory tract, as well as ocular irritation, among others [14,22]. When children are exposed to chemicals at an early stage, it can have serious and long-lasting implications. In a similar study by Rai et al. (2009), an exposure evaluation and risk assessment of acrylic acid (AA), which may be present in diapers, was done. They found that residual AA does not have any risk to human safety [23]. Xue et al. (2017) carried out a study on the determination of bisphenols in textiles and infant clothing and found that bisphenol A and bisphenol S were present in 82% and 53% of textile samples, respectively [24]. 

#### 3.2.2. pH

Since the diapers will be in direct contact with the skin when worn, it is important that the pH be controlled so that it is within the recommended limits. When the pH is not controlled and falls out of the recommended limits, it might cause irritation and itchiness to the skin. Maintaining skin pH helps in the maintenance of the skin’s overall health. When the pH is within the recommended range, particularly the mildly acidic range, it promotes the growth of bacteria such as *Staphylococcus epidermidis*, which is beneficial in the sense that it helps protect and maintain healthy skin. In general, it has been noted that skin with alkaline pH values has high bacterial counts [25]. 

Table 3 shows the pH of the DBD from high-end, medium-end, and low-end supermarkets. The pH of DBD from high-end stores followed the order H4 > H5 > H2 > H3 > H1, that from medium-end stores followed the order M6 > M7 > M5 > M2 > M3 > M8 > M9 > M10 > M1 > M4, and that from low-end stores followed the order L1 > L2 > L3 > L4 > L5. Generally, all the samples in this study had pH values in the alkaline region. The pH was found to be high in bestselling diapers and retailers’ own brands. The highest pH in DBD from high-end stores was 7.81, from medium-end stores 8.03, and from low-end stores 8.93, which was also the highest pH among all the samples studied. Only one sample (H1) out of the five samples from high-end stores was found to have a pH within the Oeko-Tex recommended limits, and 80% of them had a pH slightly above the limit. The same was also observed for the samples purchased from low-end stores. Out of the 10 samples from medium-end stores, only 3 samples had pH values within the Oeko-Tex recommended limits. Out of the 20 samples that were studied, only five had pH values within the Oeko-Tex recommended limit, implying that the probability of buying a diaper that has a pH value that is not within the Oeko-Tex recommended limits is higher than that of buying a diaper with a pH value within the Oeko-Tex recommended limit. Wirantar et al. (2019) studied the effect of diapers on skin pH and found that the pH values increase significantly in the area covered by the diaper compared to the uncovered area [13].

#### 3.2.3. pH and Formaldehyde Statistical Analysis

The group average pH of the diapers purchased from high-end stores was 7.65, medium-end stores 7.64, and low-end stores 7.99, which was the highest, and the average for all diapers was 7.7. These values show that the average pH is slightly above the recommended limit of 7.5. Figure 4a shows the pH bar graphs of the diapers for all the samples studied, showing the recommended range (orange lines) and the extent to which the pH exceeded the upper limit of the recommended range. It can be seen that there was no sample that had a pH lower than 4.

The group average concentration of formaldehyde was 14.56 mg/kg, 16.16 mg/kg, and 14.25 mg/kg for diapers purchased from high-end stores, medium-end stores, and low-end stores, respectively, and the average for all diapers was 15.28 mg/kg. Only the medium-end stores’ group average was above the average, and the lowest was from low-end store diapers. This is shown in Figure 4b, where the bar graphs for three out of five low-end (L) samples are below the orange line, which demarcates the upper limit of the recommended values. In general, most samples that exceeded the upper limit did not do so significantly, as they are not far from the orange line for both pH and formaldehyde concentration.

#### 3.2.4. Trace Metals

Heavy metals may enter the human body through the skin by absorption and may become toxic when the body does not metabolize them. Toxicity due to heavy metal absorption may result in reduced or damaged central nervous function, lungs, liver, kidneys, and other vital organs [5]. The toxicity of heavy metals is reported in the literature, and the recommended values are given by different regulations, but in this study, all comparisons were made to Oeko-Tex Standard 100. 

The concentrations of the extractable heavy metals in disposable diapers using artificial sweat are shown in Table 4, and they were found to be in the following ranges: Pb (0.002–0.019 mg/kg), Cu (0.001–0.019 mg/kg), As (0.001–0.013 mg/kg), Zn (0.014–0.644 mg/kg), Co (0.002–0.013 mg/kg), Ni (0.022–0.197 mg/kg), Cr (0.013–0.119 mg/kg), Cd (0.001–0.006 mg/kg), Mn (0.002–0.034 mg/kg), and Se (0.002–0.038 mg/kg). The concentrations of the heavy metals in all the DBD studied followed the order Zn > Ni > Cr > Pb > Se > Mn > Cu > Co = As > Cd. Although zinc had the highest concentrations in most samples compared to other elements, the levels did not exceed the Oeko-Tex recommended limits. This means that all DBD do not pose a health risk to babies due to zinc. Cadmium had the lowest maximum concentration (ca. 0.006 mg/kg) compared to other DBD samples extracted using artificial sweat. Generally, the concentrations of all the extractable heavy metals were below the Oeko-Tex recommended limits in all DBD samples except Ni, whose concentration was 0.197 mg/kg in sample H2, which was above the limit of 0.1 mg/kg. This means that all DBD in this study pose no health risk due to the heavy metals, except for Ni. Cadmium and selenium concentrations were the lowest in almost all samples.

The levels of extractible heavy metals in DBD using artificial urine are shown in Table 5, and they follow the order Zn > Ni > Cr > Se > Mn > As > Cu > Co > Pb > Cd. For the elements Zn, Ni, Cr, and Cd, the order is the same as observed when artificial sweat was used for extraction, but for other elements the order of abundance changed. This difference in concentrations could be due to the difference in the solubilities of these elements in different solutions used for extraction. Zn was the most abundant element in DBD. L2 is the only sample that had a Ni concentration (ca 0.224 mg/kg) above the Oeko-Tex recommended limits. Ni is the only heavy metal that was found to be above the recommended limits when artificial urine was used in extraction. Human exposure to nickel can cause allergies, kidney and cardiovascular diseases, as well as nasal and lung cancer [26]. Lead is one of the dangerous heavy metals that can be found in baby products, and in this study, it was below the recommended limits. In a study by Negev et al. (2018), 23% of the studied samples (baby jewelry) in Israel had a lead concentration that exceeded the US standard recommended limits [27]. In contrast, these jewelries are among the unregulated items in Israel.

## 4. Recommendations

It is important for manufacturers to have their raw materials tested for harmful substances before they can be used for manufacturing diapers to check whether such substances are within the recommended limits so as not to pose a health risk to humans. The fragrances, skin care ingredients, dyes, and other additives added to the diapers should be selected with extra care and should be minimal since they are worn on areas of the skin that are highly absorptive. The fragrances added should comply with the Code of Practice of the International Fragrance Association (IFRA) to make sure that they are not allergic. The use of fragrances, particularly those that are likely to have sensitizing effects on the skin, should be eliminated. Since the pH of most diapers in this study and the formaldehyde concentration of some of the diapers exceeded the recommended limits, it is recommended that the diaper manufacturing process be improved in order to reduce the presence of hazardous substances as much as possible. 

## 5. Conclusions

Although there are several chemicals that might be present in diapers, this study focused on pH determination, heavy metals, and formaldehyde, because an excess of any of these could be a contributing factor in causing rash in babies. Some of the chemicals that might be found in diapers can either be the result of raw-material contamination, such as pesticides, or be formed during other manufacturing processes, which include bleaching and bonding. H1, M4, and L5 had both the least formaldehyde concentrations and pH values from high-end stores, medium-end stores, and low-end stores, respectively, implying that they are the best diapers in terms of safety for the end user. Nickel is the only heavy metal that was found to exceed the Oeko-Tex recommended limits in extraction using artificial sweat (ca 0.197 mg/kg in sample H2) and artificial urine (ca 0.224 mg/kg in sample L2). The concentrations of all other elements were found to be within the recommended limits, posing no risk to babies. Only 25% of the diapers studied had pH values not within the Oeko-Tex recommended limits, whereas 50% had a formaldehyde concentration above the Oeko-Tex recommended limits.

## Figures and Tables

**Figure 1 ijerph-20-01023-f001:**
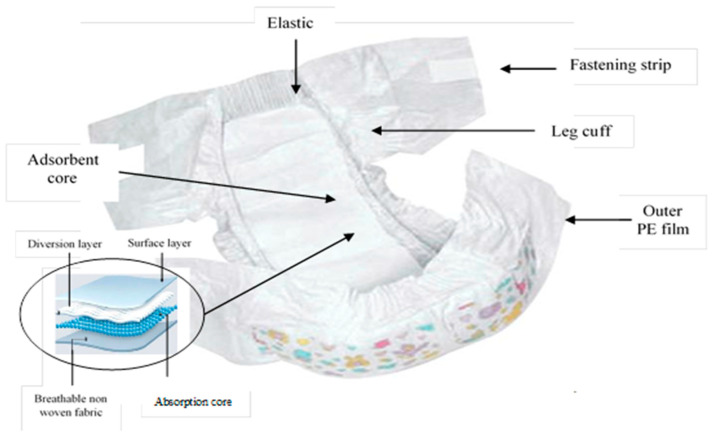
Schematic representation of a baby disposable diaper.

**Figure 2 ijerph-20-01023-f002:**
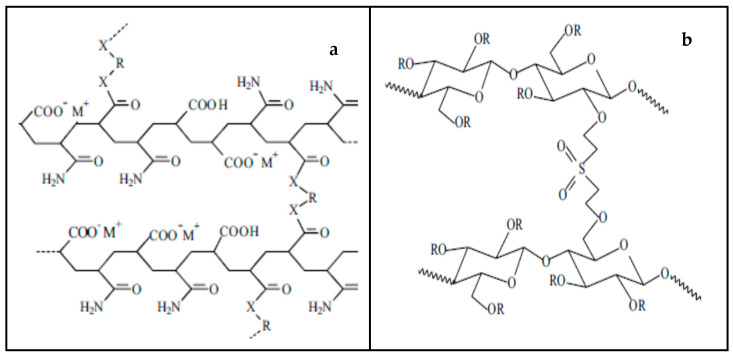
(**a**) Acrylic SAP network and (**b**) cellulose-based SAP prepared via direct cross-linking of sodium carboxymethyl cellulose (Adapted from Zohourian & Kabiri., 2008) [19].

**Figure 3 ijerph-20-01023-f003:**
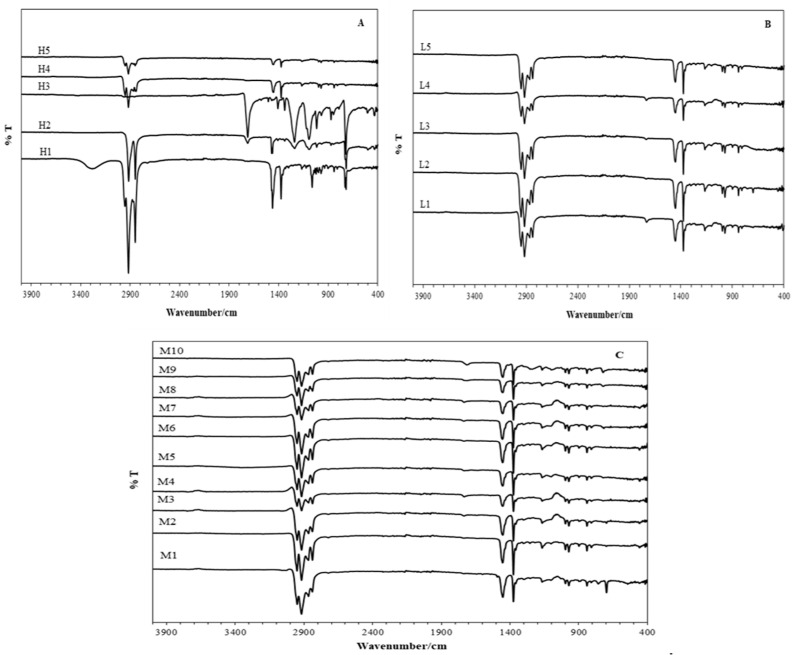
FTIR spectra of the absorbent core of (**A**) high-end diapers, (**B**) low-end diapers, and (**C**) medium-end diapers.

**Figure 4 ijerph-20-01023-f004:**
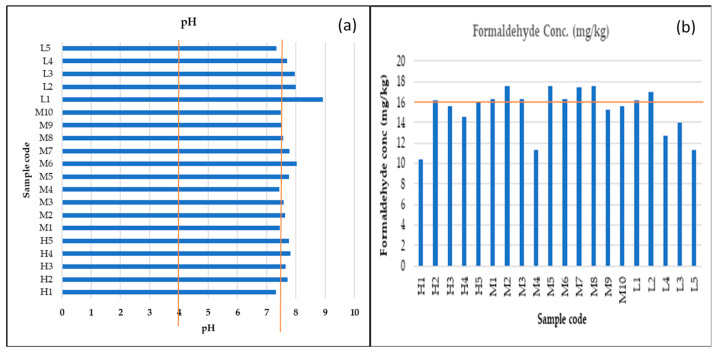
Bar graphs of (**a**) pH and (**b**) formaldehyde concentration. The orange lines indicate the recommend limits.

**Table 1 ijerph-20-01023-t001:** Sample code, description, country of manufacture, color of section in direct contact with skin of diapers from a high-end (H samples), medium-end (M samples), and low-end (L samples) market store.

Sample Code	Sample Description	Colur of the Section in Direct Contact with Skin	Made in
H1	Ingredients include petroleum, stearyl alcohol, paraffinum, liquidium, aloe barbadensis leaf extract.	Slightly blue	Poland
H2	Has wetness indicator, dual leak guard, soft cotton feel.	white	China
H3	SA material, fluff pulp, non-woven, leakage protection.	white	South Africa
H4	Ingredients include petroleum, stearyl alcohol, paraffinum, liquidium, aloe barbadensis leaf extract.	White and blue	Poland
H5	Has active channels to absorb wetness and distribute it evenly, SA core.	Slightly blue	South Africa
M1	I.N.A.	White	South Africa
M2	SA core, leak protection and breathable back sheet.	Light green	South Africa
M3	Has wetness indicator, multicore with supergel, double leakage barrier.	Purple	South Africa
M4	SA core, anti-leak guard, breathable cover.	Light green	South Africa
M5	Diamond embossed core with SA, dry guard layer, and anti-leak plastics.	White	South Africa
M6	SA gel, multicore with supergel, extra soft, wetness indicator.	Purple	South Africa
M7	SA gel, multicore with supergel, wetness indicator.	Purple	South Africa
M8	I.N.A.	Light green	South Africa
M9	SA lockgel core, stretchy waistband.	white	South Africa
M10	Two-dimensional absorbent core with side leak guard.	Light green	South Africa
L1	I.N.A.		Malaysia
L2	SA gel, dual air leakage.	Blue	South Africa
L3	I.N.A.	white	I.N.A
L4	I.N.A.	white	South Africa
L5	I.N.A.	white	I.N.A

I.N.A.: Information not available on the packaging.

**Table 2 ijerph-20-01023-t002:** Formaldehyde concentration in DBD available on the South African market and the Oeko-Tex limits.

Sample Code	Formaldehyde Conc. (mg/kg)	Recommended (mg/kg)
H1	10.438	<16 mg/kg
H2	16.205	<16 mg/kg
H3	15.616	<16 mg/kg
H4	14.557	<16 mg/kg
H5	15.969	<16 mg/kg
M1	16.322	<16 mg/kg
M2	17.617	<16 mg/kg
M3	16.322	<16 mg/kg
M4	11.380	<16 mg/kg
M5	17.617	<16 mg/kg
M6	16.322	<16 mg/kg
M7	17.499	<16 mg/kg
M8	17.617	<16 mg/kg
M9	15.263	<16 mg/kg
M10	15.616	<16 mg/kg
L1	16.205	<16 mg/kg
L2	17.029	<16 mg/kg
L3	12.674	<16 mg/kg
L4	13.969	<16 mg/kg
L5	11.380	<16 mg/kg

**Table 3 ijerph-20-01023-t003:** pH of fabrics from high-end, medium-end, and low-end stores and the Oeko-Tex recommended values.

Sample Code	pH	Recommended
H1	7.32	4.0–7.5
H2	7.71	4.0–7.5
H3	7.65	4.0–7.5
H4	7.81	4.0–7.5
H5	7.77	4.0–7.5
M1	7.46	4.0–7.5
M2	7.63	4.0–7.5
M3	7.58	4.0–7.5
M4	7.44	4.0–7.5
M5	7.76	4.0–7.5
M6	8.03	4.0–7.5
M7	7.78	4.0–7.5
M8	7.57	4.0–7.5
M9	7.52	4.0–7.5
M10	7.49	4.0–7.5
L1	8.93	4.0–7.5
L2	8.00	4.0–7.5
L3	7.96	4.0–7.5
L4	7.70	4.0–7.5
L5	7.34	4.0–7.5

**Table 4 ijerph-20-01023-t004:** Concentration of extractible heavy metals in DBD using artificial sweat (mg/kg).

Heavy Metal	Pb	Cu	As	Zn	Co	Ni	Cr	Cd	Mn	Se
H1	0.031	0.009	0.001	0.040	0.013	0.059	0.040	0.006	0.024	0.005
H2	0.016	0.011	0.013	0.644	0.002	**0.197**	0.042	0.001	0.014	0.005
H3	0.013	0.013	0.002	0.079	0.013	0.053	0.038	0.006	0.022	0.006
H4	0.045	0.005	0.002	0.252	0.013	0.056	0.119	0.006	0.018	0.005
H5	0.018	0.011	0.002	0.292	0.013	0.049	0.082	0.006	0.030	0.038
M1	0.004	0.019	0.003	0.025	0.013	0.022	0.050	0.006	0.034	0.004
M2	0.094	0.001	n.d.	0.131	0.013	0.056	0.039	0.006	0.026	0.005
M3	0.027	0.008	0.001	0.115	0.013	0.055	0.040	0.006	0.025	0.006
M4	0.056	0.010	n.d.	0.072	0.013	0.053	0.023	0.006	0.026	0.002
M5	0.022	0.012	0.001	0.093	0.013	0.052	0.041	0.006	0.027	0.005
M6	0.041	0.010	0.001	0.154	0.013	0.057	0.032	0.006	0.023	0.005
M7	0.034	0.014	0.001	0.208	0.013	0.055	0.029	0.006	0.024	0.006
M8	0.035	0.005	0.001	0.223	0.013	0.053	0.020	0.006	0.022	0.006
M9	0.023	0.014	0.001	0.167	0.013	0.050	0.028	0.006	0.024	0.005
M10	0.042	0.010	0.001	0.138	0.013	0.022	0.040	0.006	0.026	0.002
L1	0.042	0.008	0.002	0.040	0.013	0.069	0.042	0.006	0.022	0.006
L2	0.011	0.011	0.013	0.014	0.002	0.003	0.013	0.001	0.002	0.055
L3	0.002	0.011	0.002	0.087	0.013	0.060	0.024	0.006	0.027	0.006
L4	0.042	0.013	0.002	0.107	0.013	0.058	0.042	0.006	0.025	0.005
L5	0.081	0.007	0.004	0.091	0.013	0.050	0.041	0.006	0.026	0.005
Min	0.002	0.001	0.001	0.014	0.002	0.022	0.013	0.001	0.002	0.002
Max	0.094	0.019	0.013	0.644	0.013	0.197	0.119	0.006	0.034	0.038

Oeko-Tex standard 100 limits: Pb < 0.2; As < 0.2; Co < 1.0; Cr < 1.0; Ni < 0.1; Cd < 0.1, Zn < 750, Mn < 90, Cu < 25, Se < 100.

**Table 5 ijerph-20-01023-t005:** Concentration of extractible heavy metals in DBD using artificial urine (mg/kg).

Heavy Metal	Pb	Cu	As	Zn	Co	Ni	Cr	Cd	Mn	Se
H1	0.011	0.010	0.007	0.040	n.d.	0.009	0.036	0.001	0.005	0.023
H2	0.002	0.015	n.d.	0.006	0.013	0.022	0.008	0.006	0.030	0.005
H3	0.011	0.018	n.d.	0.021	0.013	0.019	0.045	0.006	0.031	0.006
H4	0.006	0.011	0.005	0.071	n.d.	0.016	0.078	0.001	0.009	0.032
H5	0.006	0.011	0.013	0.032	n.d.	0.001	0.077	0.001	0.002	0.035
M1	0.008	0.014	n.d.	0.022	0.013	0.019	0.050	0.006	0.030	0.005
M2	0.011	0.019	0.004	0.029	0.013	0.024	0.049	0.006	0.034	0.005
M3	0.007	0.013	0.001	0.005	0.013	0.004	0.045	0.006	0.030	0.005
M4	0.008	0.014	0.001	0.015	0.013	0.018	0.023	0.039	0.018	0.004
M5	0.008	0.008	0.014	0.036	n.d.	0.014	0.034	0.001	0.008	0.044
M6	0.007	0.011	0.031	0.009	0.009	0.002	0.001	0.001	0.003	0.038
M7	0.009	0.017	0.001	0.027	0.013	0.024	0.046	0.006	0.013	0.004
M8	0.009	0.015	0.001	0.011	0.013	0.020	0.028	0.006	0.030	0.006
M9	0.011	0.011	0.014	0.014	0.004	0.012	0.017	0.001	0.006	0.037
M10	0.009	0.013	n.d.	0.023	0.013	0.055	0.025	0.006	0.032	0.001
L1	0.001	0.013	0.002	0.012	0.013	0.021	0.028	0.006	0.031	0.005
L2	0.009	0.008	0.014	0.230	0.002	**0.224**	0.021	0.001	0.012	0.004
L3	0.008	0.012	0.004	0.002	n.d.	0.001	0.013	n.d.	n.d.	0.061
L4	0.008	0.014	n.d.	0.016	0.013	0.020	0.040	0.006	0.030	0.004
L5	0.010	0.013	0.002	0.014	0.013	0.018	0.041	0.006	0.031	0.004
Min	0.001	0.008	0.001	0.002	0.004	0.001	0.001	0.001	0.002	0.004
Max	0.011	0.019	0.031	0.230	0.013	0.224	0.078	0.006	0.034	0.061

Oeko-Tex standard 100 limit values: Pb < 0.2; As < 0.2; Co < 1.0; Cr < 1.0; Ni < 0.1; Cd < 0.1, Zn < 750, Mn < 90, Cu < 25, Se < 100.

## Data Availability

All the data used to support the findings of this study are included in the article.

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
