# Peer review of "Determination of Selected Harmful Substances in Baby Diapers Available on the South African Market"

_ijerph, 2023, doi:10.3390/ijerph20021023_

Round 1

Reviewer 1 Report

In the item materials and methods:

- A topic related to the statistical part of the study; for a better foundation in the data analysis.

- Mention a few more references in the discussion, as I found a minimum number of studies to support the work—present data from studies with children and blood metal exposure levels.

Author Response

The authors tried to address all the comments. Please find attached document with the responses to the comments.

Reviewer 2 Report

The authors describe the content of paraformaldehyde and heavy metals in disposable diapers found on the South African market. The introduction should include information on whether there are such studies in the available literature. In my opinion, the results and discussion chapters should be written separately. In addition, the authors should provide data on the permissible standards for the content of heavy metals in products intended for infants, especially those in direct contact with the skin.

Author Response

The authors tried to address all the comments and suggestions. Attached is the document with the responses.

Round 2

Reviewer 2 Report

After taking into account the reviewer's comments. The manuscript is suitable for publication